# NATURAL LANGUAGE INFERENCE WITH EXTERNAL KNOWLEDGE

## ABSTRACT

Modeling informal inference in natural language is very challenging. With the recent availability of large annotated data, it has become feasible to train complex models such as neural networks to perform natural language inference (NLI), which have achieved state-of-the-art performance. Although there exist relatively large annotated data, can machines learn all knowledge needed to perform NLI from the data? If not, how can NLI models benefit from external knowledge and how to build NLI models to leverage it? In this paper, we aim to answer these questions by enriching the state-of-the-art neural natural language inference models with external knowledge. We demonstrate that the proposed models with external knowledge further improve the state of the art on the Stanford Natural Language Inference (SNLI) dataset.

## 1 INTRODUCTION

Reasoning and inference are central to both human and artificial intelligence. Natural language inference (NLI) is concerned with determining whether a natural-language hypothesis $h$ can be inferred from a natural-language premise $p$. Modeling inference in human language is very challenging but is a basic problem towards true natural language understanding — NLI is regarded as a necessary (if not sufficient) condition for true natural language understanding (MacCartney & Manning, 2007).

The most recent years have seen advances in modeling natural language inference. An important contribution is the creation of much larger annotated datasets such as SNLI (Bowman et al., 2015) and MultiNLI (Williams et al., 2017). This makes it feasible to train more complex inference models. Neural network models, which often need relatively large amounts of annotated data to estimate their parameters, have shown to achieve the state of the art on SNLI and MultiNLI (Bowman et al., 2015; 2016; Munkhdalai & Yu, 2016b; Parikh et al., 2016; Sha et al., 2016; Paria et al., 2016; Chen et al., 2017a;b).

While these neural networks have shown to be very effective in estimating the underlying inference functions by leveraging large training data to achieve the best results, they have focused on end-to-end training, where all inference knowledge is assumed to be learnable from the provided training data. In this paper, we relax this assumption, by exploring whether external knowledge can further help the best reported models, for which we propose models to leverage external knowledge in major components of NLI. Consider an example from the SNLI dataset:

- $p$: An African person standing in a *wheat* field.
- $h$: A person standing in a *corn* field.

If the machine cannot learn useful or plenty information to distinguish the relationship between *wheat* and *corn* from the large annotated data, it is difficult for a model to predict that the premise contradicts the hypothesis.

In this paper, we propose neural network-based NLI models that can benefit from external knowledge. Although in many tasks learning *tabula rasa* achieved state-of-the-art performance, we believe complicated NLP problems such as NLI would benefit from leveraging knowledge accumulated by humans, at least in a foreseeable future when machines are unable to learn that with limited data.

A typical neural-network-based NLI model consists of roughly four components — encoding the input sentences, performing co-attention across premise and hypothesis, collecting and computing local inference, and performing sentence-level inference judgment by aggregating or composing local information information. In this paper, we propose models that are capable of leveraging external knowledge in co-attention, local inference collection, and inference composition components. We demonstrate that utilizing external knowledge in neural network models outperforms the previously reported best models. The advantage of using external knowledge is more significant when the size of training data is restricted, suggesting that if more knowledge can be obtained, it may yielding more benefit. Specifically, this study shows that external semantic knowledge helps mostly in attaining more accurate local inference information, but also benefits co-attention and aggregation of local inference.

## 2    RELATED WORK

Early work on natural language inference (also called recognizing textual textual) has been performed on quite small datasets with conventional methods, such as shallow methods (Glickman et al., 2005), natural logic methods (MacCartney & Manning, 2007), among others. These work already shows the usefulness of external knowledge, such as WordNet (Miller, 1995), FrameNet (Baker et al., 1998), and so on.

More recently, the large-scale dataset SNLI was made available, which made it possible to train more complicated neural networks. These models fall into two kind of approaches: sentence encoding-based models and inter-sentence attention-based models. Sentence encoding-based models use Siamese architecture (Bromley et al., 1993) — the parameter-tied neural networks are applied to encode both the premise and hypothesis. Then a neural network classifier (i.e., multilayer perceptron) is applied to decide the relationship between the two sentence representations. Different neural networks have been utilized as sentence encoders, such as LSTM (Bowman et al., 2015), GRU (Vendrov et al., 2015), CNN (Mou et al., 2016), BiLSTM and its variants (Liu et al., 2016; Lin et al., 2017; Chen et al., 2017b), and more complicated neural networks (Bowman et al., 2016; Munkhdalai & Yu, 2016a;b). The advantage of encoding-based models is that the encoders transform sentences into fixed-length vector representations, which can help a wide range of transfer tasks (Conneau et al., 2017). However, this architecture ignores the local interaction between two sentences, which is necessary in traditional natural language inference procedure (MacCartney & Manning, 2007).

Therefore, inter-sentence attention-based models were proposed to relieve this problem. In this framework, local inference information is collected by the attention mechanism and then fed into neural networks to compose as fixed-sized vectors before the final classification. Many related works follow this route (Rocktäschel et al., 2015; Wang & Jiang, 2016; Cheng et al., 2016; Parikh et al., 2016; Chen et al., 2017a). Among them, Rocktäschel et al. (2015) were the first to propose neural attention-based models for NLI. Chen et al. (2017a) proposed an enhanced sequential inference model (ESIM), which is one of the best models so far and regarded as the baseline in this paper.

In general, external knowledge have been shown to be effective in a wide range of NLP tasks, including machine translation (Shi et al., 2016; Zhang et al., 2017), language modeling (Ahn et al., 2016), and dialogue system (Chen et al., 2016). For NLI, to the best of our knowledge, we are the first to utilize external knowledge together with neural networks. In this paper, we first show that a neural network equipped with external knowledge obtains further improvement over the already strong baseline, and achieves an accuracy of 88.6% on the SNLI benchmark. Furthermore, we show that the gain is more significant when using less training samples.

## 3    METHODS

### 3.1    REPRESENTATION OF EXTERNAL KNOWLEDGE

External knowledge needs to be converted to a numerical representation for enriching natural language inference model. One of approaches to represent external knowledge is using knowledge graph embeddings, such as TransE (Bordes et al., 2013), TransH (Wang et al., 2014), TransG (Xiao et al., 2016), and so on. However, these kind of approaches usually need to train a knowledge-graph embedding beforehand.

In this paper, we propose to use relation features to describe relationship between the words in any word pair, which can be easily obtained from various knowledge graphs, such as WordNet (Miller, 1995), and Freebase (Bollacker et al., 2008). Specifically, we use WordNet to measure the semantic relatedness of the word in a pair using various relation types, including synonymy, antonymy, hypernymy, and so on. Each of these features is a real number on the interval [0,1]. The definition and instances of pair features derived from WordNet are indicated in Table 1. The setting of features refers to MacCartney (2009), but we add a new feature *same hypernym*, which improve the result significantly in our experiments. Intuitively, the *synonymy*, *hypernymy* and *hyponymy* features help model the entailment of word pairs; the *antonymy* and *same hypernym* features help model contradiction in word pairs.

We regard the vector $r \in \mathbb{R}^{D_r}$ as the relation feature derived from external knowledge, where $D_r$ is 5 in our experiments. The $r$ will be enriched in the neural inference model to capture external semantic knowledge. Table 2 reports some key statistics of the relation features from WordNet.

Table 1: The definition and instances of relation features from WordNet

| TYPE | DEFINITION | INSTANCES |
|------|------------|-----------|
| *Synonymy* | It takes the value $1$ if the words in the pair are synonyms in WordNet (i.e., belong to the same synset), and $0$ otherwise. Specifically, it takes value 1 when two words are same. | [felicitous, good] = 1 [dog, wolf] = 0 |
| *Antonymy* | It takes the value $1$ if the words in the pair are antonyms in WordNet, and $0$ otherwise. | [wet, dry] = 1 |
| *Hypernymy* | It takes the value $1-n/8$ if one word is a (direct or indirect) hypernym of the other word in WordNet, where $n$ is the number of edges between the two words in hierarchies, and $0$ otherwise. | [dog, canid] = 0.875 [wolf, canid] = 0.875 [dog, carnivore] = 0.75 [canid, dog] = 0 |
| *Hyponymy* | It takes the value $1 - n/8$ if a word is a (direct or indirect) hyponym of the other word in WordNet, where $n$ is the number of edges between the two words in hierarchies, and $0$ otherwise. | [canid, dog] = 0.875 [canid, wolf] = 0.875 [carnivore, dog] = 0.75 [dog, canid] = 0 |
| *Same Hypernym* | It takes the value $1$ if the two words have the same hypernym but they do not belong to the same synset, and $0$ otherwise. | [dog, wolf] = 1 |

Table 2: The key statistics of relation features from WordNet.

| TYPE | #WORDS | #PAIRS |
|------|--------|--------|
| *Synonymy* | 84,487 | 237,937 |
| *Antonymy* | 6,161 | 6,617 |
| *Hypernymy* | 57,475 | 753,086 |
| *Hyponymy* | 57,475 | 753,086 |
| *Same Hypernym* | 53,281 | 3,674,700 |

## 3.2 NEURAL INFERENCE MODELS

We present here our natural inference models which are composed of the following major components: input encoding, knowledge enriched co-attention, knowledge enriched local inference collection, and knowledge enriched inference composition. Figure 1 shows a high-level view of the

architecture. First, the premise and hypothesis are encoded by the input encoding components as context-dependent representations. Second, co-attention is calculated to obtain word-level soft-alignment between two sentences. Third, local inference information is collected to prepare for final prediction. Fourth, the inference composition component applies aggregation of the whole sentences and makes final prediction based on the fixed-size vector. Among them, external knowledge is regard as the auxiliary component to improve the ability of (1) calculating co-attention, (2) collecting local inference information and (3) composing inference.

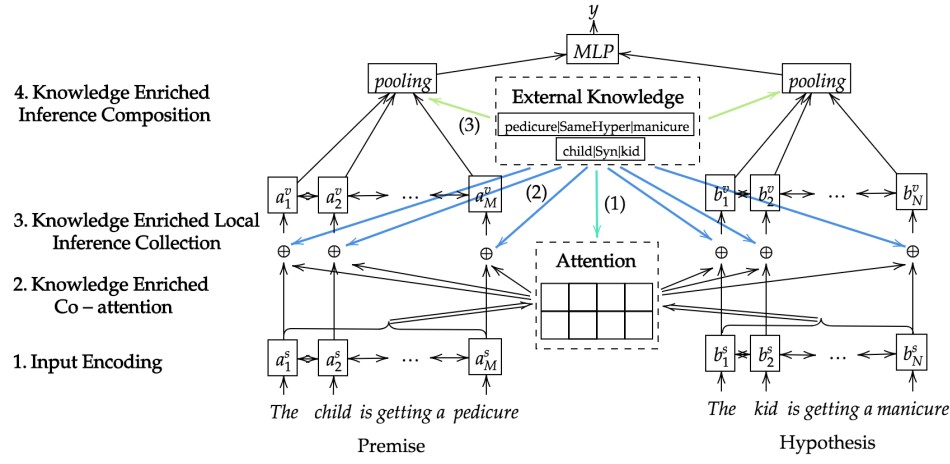

Figure 1: A high-level view of our neural inference networks. Given two sentence, i.e., the premise *"The child is getting a pedicure"*, and the hypothesis *"The kid is getting a manicure"*, the model needs to predict the relationship among them: *entailment*, *contradiction*, or *neutral*.

### 3.2.1 INPUT ENCODING

Given the word sequences of the premise $\boldsymbol{a} = (a_1, \ldots, a_M)$ and the hypothesis $\boldsymbol{b} = (b_1, \ldots, b_N)$, where $M$ and $N$ are the lengths of the sentences, the final objective is to predict a label $y$ that indicates the logic relationship between $\boldsymbol{a}$ and $\boldsymbol{b}$. The formula is

$$y^\star = \arg\max_y P_r(y|\boldsymbol{a}, \boldsymbol{b}),\tag{1}$$

Specifically, "<BOS>" and "<EOS>" are inserted as the first and last token, respectively. First, $\boldsymbol{a}$ and $\boldsymbol{b}$ are embedded into a $D_e$-dimensional vectors $[\mathbf{E}(a_1), \ldots, \mathbf{E}(a_M)]$ and $[\mathbf{E}(b_1), \ldots, \mathbf{E}(b_N)]$ using an embedding matrix $\mathbf{E} \in \mathbb{R}^{D_e \times V}$, where $V$ is the vocabulary size and $\mathbf{E}$ can be initialized with some pre-trained word embeddings from a universal corpus.

To represent the words of the premise and hypothesis in a context-dependent way, the two sentences are fed into the encoders to obtain context-dependent hidden states $\boldsymbol{a}^s$ and $\boldsymbol{b}^s$. The formula is

$$\boldsymbol{a}_i^s = \text{Encoder}(\mathbf{E}(\boldsymbol{a}), i), \ \boldsymbol{b}_j^s = \text{Encoder}(\mathbf{E}(\boldsymbol{b}), j).\tag{2}$$

We employ bidirectional LSTMs (BiLSTMs) (Hochreiter & Schmidhuber, 1997) as encoders, which is a common choice for natural language. A BiLSTM runs a forward and a backward LSTM on a sequence starting from the left and the right end, respectively. The hidden states generated by these two LSTMs at each time step are concatenated to represent that time step and its context: $\boldsymbol{h}_t = [\boldsymbol{h}_t^{\rightarrow}; \boldsymbol{h}_t^{\leftarrow}]$. The hidden states of the unidirectional LSTM ($\boldsymbol{h}_t^{\rightarrow}$ or $\boldsymbol{h}_t^{\leftarrow}$) is calculated as follows:

$$\boldsymbol{i}_t = \sigma(\mathbf{W}_i\boldsymbol{x}_t + \mathbf{U}_i\boldsymbol{h}_{t-1} + \boldsymbol{b}_i),\tag{3}$$

$$\boldsymbol{f}_t = \sigma(\mathbf{W}_f\boldsymbol{x}_t + \mathbf{U}_f\boldsymbol{h}_{t-1} + \boldsymbol{b}_f),\tag{4}$$

$$\boldsymbol{u}_t = \tanh(\mathbf{W}_u\boldsymbol{x}_t + \mathbf{U}_u\boldsymbol{h}_{t-1} + \boldsymbol{b}_u),\tag{5}$$

$$\boldsymbol{o}_t = \sigma(\mathbf{W}_o\boldsymbol{x}_t + \mathbf{U}_o\boldsymbol{h}_{t-1} + \boldsymbol{b}_o),\tag{6}$$

$$\boldsymbol{c}_t = \boldsymbol{f}_t \odot \boldsymbol{c}_{t-1} + \boldsymbol{i}_t \odot \boldsymbol{u}_t,\tag{7}$$

$$\boldsymbol{h}_t = \boldsymbol{o}_t \odot \tanh(\boldsymbol{c}_t),\tag{8}$$

where $\sigma$ is the sigmoid function, $\odot$ is the element-wise multiplication of two vectors. $\mathbf{W}_i, \mathbf{W}_f, \mathbf{W}_u, \mathbf{W}_o \in \mathbb{R}^{D \times D_e}$, $\mathbf{U}_i, \mathbf{U}_f, \mathbf{U}_u, \mathbf{U}_o \in \mathbb{R}^{D \times D}$ and $\boldsymbol{b}_i, \boldsymbol{b}_f, \boldsymbol{b}_u, \boldsymbol{b}_o \in \mathbb{R}^D$ are parameters to be learned. $D$ is the dimension of the hidden states in the LSTM. The LSTM utilizes a set of gating functions for each input vector $\boldsymbol{x}_t$, i.e., the input gate $\boldsymbol{i}_t$, forget gate $\boldsymbol{f}_t$, and output gate $\boldsymbol{o}_t$, together with a memory cell $\boldsymbol{c}_t$ to generate a hidden state $\boldsymbol{h}_t$.

### 3.2.2 KNOWLEDGE-ENRICHED CO-ATTENTION

In this component, we acquire soft-alignment of word pairs between the premise and hypothesis based on our knowledge-enriched co-attention mechanism. Given the relation features $\boldsymbol{r}_{ij} \in \mathbb{R}^{D_r}$ between the premise's $i$-th word and the hypothesis's $j$-th word from the external knowledge, the co-attention is calculated as

$$e_{ij} = (\boldsymbol{a}_i^s)^{\mathrm{T}} \boldsymbol{b}_j^s + F(\boldsymbol{r}_{ij}) . \tag{9}$$

The function $F$ can be any non-linear or linear function. Here we use $F(\boldsymbol{r}_{ij}) = \lambda \mathbb{1}(\boldsymbol{r}_{ij})$, where $\lambda$ is a hyper-parameter tuned on the development set and $\mathbb{1}$ is the indication function.

$$\mathbb{1}(\boldsymbol{r}_{ij}) = \begin{cases} 1 & \text{if } \boldsymbol{r}_{ij} \text{ is not zero vector} ; \\ 0 & \text{if } \boldsymbol{r}_{ij} \text{ is zero vector} . \end{cases} \tag{10}$$

Intuitively, the word pairs with semantic relationship in various features are probably aligned together. Soft-alignment is determined by the co-attention matrix $\mathbf{e} \in \mathbb{R}^{M \times N}$ computed in Equation (9), which is used to obtain the local relevance between the premise and hypothesis. For the hidden state of a word in a premise, i.e., $\boldsymbol{a}_i^s$ (already encoding the word itself and its context), the relevant semantics in the hypothesis is identified into a context vector $\boldsymbol{a}_i^c$ using $e_{ij}$, more specifically with Equation (11).

$$\alpha_{ij} = \frac{\exp(e_{ij})}{\sum_{k=1}^N \exp(e_{ik})} , \; \boldsymbol{a}_i^c = \sum_{j=1}^N \alpha_{ij} \boldsymbol{b}_j^s , \tag{11}$$

$$\beta_{ij} = \frac{\exp(e_{ij})}{\sum_{k=1}^M \exp(e_{kj})} , \; \boldsymbol{b}_j^c = \sum_{i=1}^M \beta_{ij} \boldsymbol{a}_i^s , \tag{12}$$

where $\boldsymbol{\alpha} \in \mathbb{R}^{M \times N}$ and $\boldsymbol{\beta} \in \mathbb{R}^{M \times N}$ are the normalized attention weight matrices with respect to the 2-axis and 1-axis. The same calculation is performed for each word in the hypothesis, i.e., $\boldsymbol{b}_j^s$, with Equation (12) to obtain the context vector $\boldsymbol{b}_j^c$.

### 3.2.3 KNOWLEDGE-ENRICHED LOCAL INFERENCE COLLECTION

By way of comparing the relationship between $\boldsymbol{a}^s$ and $\boldsymbol{a}^c$ (or $\boldsymbol{b}^s$ and $\boldsymbol{b}^c$), we can model local inference between aligned word pairs. In this component, we further collect knowledge-enriched local inference information. The formula is

$$\boldsymbol{a}_i^m = G([\boldsymbol{a}_i^s; \boldsymbol{a}_i^c; \boldsymbol{a}_i^s - \boldsymbol{a}_i^c; \boldsymbol{a}_i^s \odot \boldsymbol{a}_i^c; \sum_{j=1}^N \alpha_{ij} \boldsymbol{r}_{ij}]) , \tag{13}$$

$$\boldsymbol{b}_j^m = G([\boldsymbol{b}_j^s; \boldsymbol{b}_j^c; \boldsymbol{b}_j^s - \boldsymbol{b}_j^c; \boldsymbol{b}_j^s \odot \boldsymbol{b}_j^c; \sum_{i=1}^M \beta_{ij} \boldsymbol{r}_{ji}]) , \tag{14}$$

where a heuristic matching trick with difference and element-wise product is used (Mou et al., 2016; Chen et al., 2017a). The last term in Equation (13)(14) aims to obtain the local inference relationship between the original vectors ($\boldsymbol{a}_i^s$ or $\boldsymbol{b}_j^s$) and the context vectors ($\boldsymbol{a}_i^c$, or $\boldsymbol{b}_j^c$) derived from the external semantic knowledge $\boldsymbol{r}_{ij}$. $G$ is a non-linear mapping function to reduce dimensionality. Specifically, we use a 1-layer feed-forward neural network with the ReLU activation function with a shortcut connection from input $\sum_{j=1}^N \alpha_{ij} \boldsymbol{r}_{ij}$ (and $\sum_{i=1}^M \beta_{ij} \boldsymbol{r}_{ji}$). Intuitively, we use a weighted version of the relation vectors between the premise and hypothesis, because only semantic relations of aligned word pairs make an important impact on the whole sentence inference relationship.

### 3.2.4 Knowledge-Enriched Inference Composition

In this component, we introduce knowledge-enriched inference composition. To determine the overall inference relationship between a premise and a hypothesis, we need to explore a composition layer to compose the local inference vectors ($\boldsymbol{a}^m$ and $\boldsymbol{b}^m$) collected above. The formula is

$$\boldsymbol{a}_i^v = \text{Composition}(\boldsymbol{a}^m, i)\,,\ \boldsymbol{b}_j^v = \text{Composition}(\boldsymbol{b}^m, j)\,. \tag{15}$$

Here, we also use BiLSTMs as building blocks for the composition layer. The BiLSTMs read local inference vectors ($\boldsymbol{a}^m$ and $\boldsymbol{b}^m$) and learn to judge the type of local inference relationship and distinguish crucial local inference vectors for overall sentence-level inference relationship. The responsibility of BiLSTMs in the inference composition layer is completely different from the BiLSTMs in the input encoding layer. Our inference model converts the output hidden vectors of BiLSTMs to a fixed-length vector with pooling operations and puts it into the final classifier to determine the overall inference class. Particularly, besides using mean pooling and max pooling similarly to ESIM (Chen et al., 2017a), we propose to use weighted pooling based on external knowledge to obtain a fixed-length vector as in Equation (16). Intuitively, the final prediction is mostly determined by those word pairs appearing in the external knowledge. Chen et al. (2017b) uses a similar idea called gated-attention but they do not use external knowledge.

$$\boldsymbol{a}^{\text{w}} = \sum_{i=1}^M \frac{\exp(H(\sum_{j=1}^N \alpha_{ij}\boldsymbol{r}_{ij}))}{\sum_{i=1}^M \exp(H(\sum_{j=1}^N \alpha_{ij}\boldsymbol{r}_{ij}))}\boldsymbol{a}_i^v\,,\ \boldsymbol{b}^{\text{w}} = \sum_{j=1}^N \frac{\exp(H(\sum_{i=1}^M \beta_{ij}\boldsymbol{r}_{ji}))}{\sum_{j=1}^N \exp(H(\sum_{i=1}^M \beta_{ij}\boldsymbol{r}_{ji}))}\boldsymbol{b}_j^v\,. \tag{16}$$

In our experiments, we regard the function $H$ as a 1-layer feed-forward neural network with ReLU activation function. We concatenate all pooling vectors, i.e., mean, max, and weighted pooling, into a fixed-length vector and then put the vector into a final multilayer perceptron (MLP) classifier. The MLP has a hidden layer with *tanh* activation and *softmax* output layer in our experiments. The entire model is trained end-to-end, through minimizing the cross-entropy loss.

## 4 Experimental Setup

### 4.1 Data

The Stanford Natural Language Inference (SNLI) dataset (Bowman et al., 2015) focuses on three basic relationships between a premise and a potential hypothesis: the premise entails the hypothesis (*entailment*), they contradict each other (*contradiction*), or they are not related (*neutral*). We use the same data split as in previous work, and use classification accuracy as the evaluation metric, as in related work. WordNet 3.0 (Miller, 1995) is used to extract semantic relation features between words, as described in Section 3.1. The words are lemmatized using Stanford CoreNLP 3.7.0 (Manning et al., 2014) to match words in WordNet, but the input word sequences for the input encoding layer are only tokenized, without lemmatization.

### 4.2 Training

We release our code at [xxx] to make it replicatibility purposes. The models are selected on the development set. Some of our training details are as follows: the dimension of the hidden states of LSTMs and word embeddings are 300. The word embeddings are initialized by *300D GloVe 840B* (Pennington et al., 2014), and out-of-vocabulary words among them are initialized randomly. All word embeddings are updated during training. Adam (Kingma & Ba, 2014) is used for optimization with an initial learning rate 0.0004. The mini-batch size is set to 32. Dropout with a keep rate of 0.5 and early stopping with patience of 7 are used to avoid overfitting. The gradient is clipped with a maximum L2-norm 10. The trade-off $\lambda$ for calculating co-attention in Equation (9) is selected in [0.1, 0.2, 0.5, 1, 2, 5, 10, 20, 50] based on the development set.

Table 3: Accuracies of the different models on SNLI. The proposed model KIM achieves an accuracy of 88.6% on the test set, which is the best result so far.

| MODEL | TEST |
|---|---|
| LSTM (Bowman et al., 2015) | 80.6 |
| GRU (Vendrov et al., 2015) | 81.4 |
| Tree CNN (Mou et al., 2016) | 82.1 |
| SPINN-PI (Bowman et al., 2016) | 83.2 |
| NTI (Munkhdalai & Yu, 2016b) | 83.4 |
| Intra-Att BiLSTM (Liu et al., 2016) | 84.2 |
| Self-Att BiLSTM (Lin et al., 2017) | 84.2 |
| NSE (Munkhdalai & Yu, 2016a) | 84.6 |
| Gated-Att BiLSTM (Chen et al., 2017b) | 85.5 |
| DiSAN (Shen et al., 2017) | 85.6 |
| LSTM Att (Rocktäschel et al., 2015) | 83.5 |
| mLSTM (Wang & Jiang, 2016) | 86.1 |
| LSTMN (Cheng et al., 2016) | 86.3 |
| Decomposable Att (Parikh et al., 2016) | 86.8 |
| NTI (Munkhdalai & Yu, 2016b) | 87.3 |
| Re-read LSTM (Sha et al., 2016) | 87.5 |
| BiMPM (Wang et al., 2017) | 87.5 |
| btree-LSTM (Paria et al., 2016) | 87.6 |
| DIM (Gong et al., 2017) | 88.0 |
| ESIM (Chen et al., 2017a) | 88.0 |
| **KIM** | **88.6** |
| HIM (ESIM+Syntactic TreeLSTM) (Chen et al., 2017a) | 88.6 |
| BiMPM (Ensemble) (Wang et al., 2017) | 88.8 |
| DIIN (Ensemble) (Wang et al., 2017) | 88.9 |
| **KIM (Ensemble)** | **89.1** |

# 5 RESULTS

## 5.1 OVERALL PERFORMANCE

Table 3 shows the results of different models on the SNLI dataset. The first group of models use sentence-encoding based approaches. Bowman et al. (2015) employs LSTMs as encoders for both the premise and hypothesis into two fixed-size sentence vectors. Then the sentence representation is put into a MLP classifier to predict the final inference relationship. The accuracy on the test set is 80.6%. Many related works follow this framework, using different neural networks as encoders. Their performances are also listed in the first group in Table 3. Among them, gated-Att BiLSTM (Chen et al., 2017b) achieves an accuracy of 85.5%, which is state of the art for sentence-encoding based approaches.

The second group of models uses a cross-sentence attention mechanism, which can obtain soft-alignment information between cross-sentence word pairs. Wang & Jiang (2016) proposes a matching-LSTM to compare the inference information of locally-aligned words, and obtains a higher accuracy of 86.1%, even better than the state-of-the-art sentence-encoding models. Other related models are also listed in the second group in Table 3. Among them, ESIM (Chen et al., 2017a) is the previous state-of-the-art system, whose accuracy in test set is 88.0%. The proposed model, namely Knowledge-based Inference Model (**KIM**), which enriches ESIM with external knowledge, obtains an accuracy of 88.6%. The difference between ESIM and KIM is statistically significant under the one-tailed paired $t$-test at the 99% significance level. To be best of our knowledge, this is a new state of the art. Our ensemble model, which averages the probability distributions from ten individual single KIMs with different initialization, achieves an even higher accuracy, 89.1%.

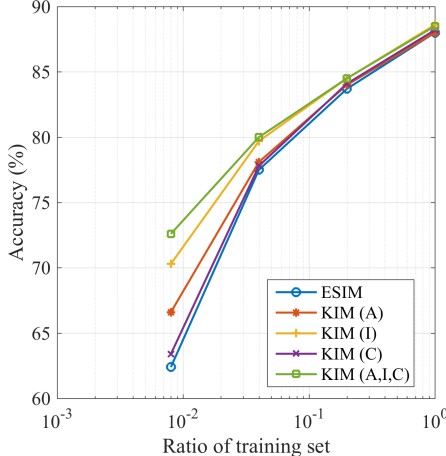 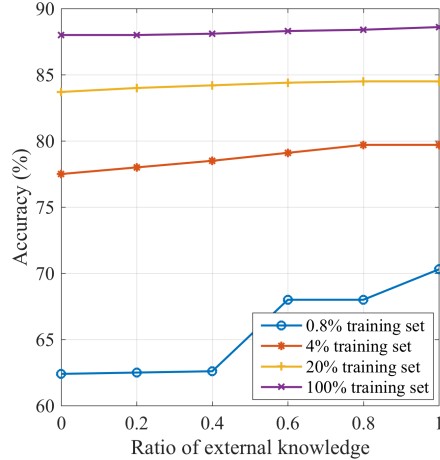

Figure 2: Accuracies of the models using different components of external knowledge, varies the training set size among 0.8%, 4%, 20% and the whole training set.

Figure 3: Accuracies of the models using different size of external knowledge. More external knowledge, higher accuracies.

## 5.2 ABLATION ANALYSIS

Figure 2 displays the ablation analysis of different components when using the external knowledge. To compare the importance of external knowledge under different training data scales, we randomly sample different ratio of the whole training set, i.e., 0.8%, 4%, 20% and 100%. "A" indicates adding external knowledge in calculating the co-attention matrix as in Equation (9), "I" indicates adding external knowledge in collecting local inference information as in Equation (13)(14), and "C" indicates adding external knowledge in composing inference as in Equation (16). When we only have restricted training data, i.e., 0.8% training set (about 4,000 samples), our baseline ESIM has a poor accuracy of 62.4%. When we only add external knowledge in calculating co-attention ("A"), the accuracy increases to 66.6% (+ absolute 4.2%). When we only utilize external knowledge in collecting local inference information ("I"), the accuracy has a significant gain, to 70.3% (+ absolute 7.9%). When we only add external knowledge in inference composition ("C"), the accuracy gets a smaller gain to 63.4% (+ absolute 1.0%). The comparison indicates that "I" plays the most important role among the three components in using external knowledge. Moreover, when we compose the three components ("A,I,C"), we obtain the best result of 72.6% (+ absolute 10.2%). When we use more training data, i.e., 4%, 20%, 100% of the training set, only utilizing external knowledge in local inference information collected ("I") achieves a significant gain, but "A" or "C" do not bring any significant improvement. The results indicate that external semantic knowledge only helps in co-attention and composition when there is limited training data, but always helps in collecting local inference information. Meanwhile, for less training data, $\lambda$ is usually set to a larger value. For example, the optimal $\lambda$ tuned on the development set is 20 for 0.8% training set, 2 for the 4% training set, 1 for the 20% training set and 0.2 for the 100% training set.

Figure 3 displays the results of using different ratio of external knowledge for different training data size. Note that here we only use external knowledge in collecting local inference information, because it always works well for different scale of the training set. Better accuracies are achieved when using more external knowledge. Especially under the condition of restricted training data (0.8%), the model obtains a large gain when using more than half of the external knowledge.

## 6 CONCLUSIONS

Our enriched neural network-based model for natural language inference with external knowledge, namely KIM, achieves a new state-of-the-art accuracy on the SNLI dataset. The model is equipped with external knowledge in the major informal inference components, specifically, in calculating

co-attention, collecting local inference, and composing inference. The proposed models of infusing neural networks with external knowledge may also help shed some light on tasks other than NLI, such as question answering and machine translation.

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
