# OpenReview forum: "Natural Language Inference with External Knowledge"
_ICLR.cc/2018/Conference — Invite to Workshop Track_

### Official Review · AnonReviewer2 · 2017-11-19
**Interesting direction; modest positive results**

**Rating:** 6
**Confidence:** 5

**Review:**

Update:

The response addressed all my major concerns, and I think the paper is sound. (I'm updating my confidence to a 5.) So, the paper makes an empirically *very* small step in an interesting line of language understanding research. This paper should be published in some form, but my low-ish score is due simply to my worry that ICLR is not the venue. I think this would be a clear 'accept' as a *ACL short paper, and would probably be viable as a *ACL long paper, but it will definitely have less impact on the overall field of representation learning than will the typical ICLR paper, so I can recommend it only with reservations.

--

This paper presents a method to use external lexical knowledge (word–word relations from WordNet) as an auxiliary input when solving the problem of textual entailment (aka NLI). The idea of accessing outside commonsense knowledge within an end-to-end trained model is one that I expect to be increasingly important in work on language understanding. This paper does not make that much progress on the problem in general—the methods here are quite specific to words and to NLI—and the proposed methods yields only yield large empirical gains in a reduced-data setting, but the paper serves as a well-executed proof of concept. In short, the paper is likely to be of low technical impact, but it's interesting and thorough enough that I lean slightly toward acceptance.

My only concern is on fair comparison: Numbers from this model are compared with numbers from the published ESIM model in several places (Table 3, Figure 2, etc.) as a way to provide evidence for the paper's core claim (that the added knowledge in the proposed model helps). This only constitutes clear evidence if the proposed model is identical to ESIM in all of its unimportant details—word representations, hyperparameter tuning methods, etc. Can the authors comment on this?

For what it's worth, the existence of another paper submission on roughly the same topic with roughly the same results (https://openreview.net/pdf?id=B1twdMCab) makes more confident that the main results in this paper are sound, since they've already been replicated, at least to a coarse approximation.

Minor points:

For TransE, what does this mean:"However, these kind of approaches usually need to train a knowledge-graph embedding beforehand."

You should say more about why you chose the constant 8 in Table 1 (both why you chose to hard code a value, and why that value).

There's a mysterious box above the text 'Figure 1'. Possibly a figure rendering error?

The LSTM equations are quite widely known. I'd encourage you to cite a relevant source and remove them.

Say more about why you choose equation (9). This notably treats all five relation types equally, which seems like a somewhat extreme simplifying assumption.

Equation (15) is confusing. Is a^m a matrix, since it doesn't have an index on it?

What is "early stopping with patience of 7"? Is that meant to mean 7 epochs?

The opening paragraph of 5.1 seems entirely irrelevant, as do the associated results in the results table. I suspect that this might be an opportunity for a gratuitous self-citation.

There are plenty of typos: "... to make it replicatibility purposes."; "Specifically, we use WordNet to measure the semantic relatedness of the *word* in a pair"; etc.

---

> ### Author Response · Authors · 2018-01-05
> **Author Responses**
>
> >>> Reviewer’s comment 1: ”You should say more about why you chose the constant 8 in Table 1 (both why you chose to hard code a value, and why that value).”
>
> Author response: Thanks. The original WordNet hierarchy is bounded by the depth of 20. In our paper, we take the setup specified in MacCartney (2009) to bound the depth at 8 (i.e., ignoring pairs in the hierarchy which have more than 8 edges in between).  Similarly, we follow MacCartney (2009) for hypernym and hyponym feature design, which we are shown in our experiments to help improve neural-network-based inference model. We will make this clearer in our revision.
>
> >>> Reviewer’s comment 2:”This only constitutes clear evidence if the proposed model is identical to ESIM in all of its unimportant details—word representations, hyperparameter tuning methods, etc. Can the authors comment on this?”
>
> Author response: We are sure that all of its unimportant details, such as hyperparameter tuning methods, are identical to ESIM. We modified the ESIM code from GitHub (https://github.com/lukecq1231/nli) and we only modified NLI components to explore external knowledge. In addition, ESIM is a well-tuned model already and tuning ESIM by itself does not yield further gain.
>
> >>> Reviewer’s comment 3: ”Say more about why you choose equation (9). This notably treats all five relation types equally, which seems like a somewhat extreme simplifying assumption.”
>
> Author response: Good comment. We actually added a MLP layer in Theano to learn the underlying combination function but did not actually observe further improvement over our best performance. We will add the results and some discussion in our revision.
>
> >>> Reviewer comments 4: What is "early stopping with patience of 7"? Is that meant to mean 7 epochs?
>
> Author response: Yes, early stopping with patience of 7 means 7 epochs. We will make this clearer.
>
> In general, as informal reasoning is a core problem, our SoTa results on one of the widely used benchmarks (SNLI), our investigation of missing knowledge in all three major inference components, as well as the analysis, in our opinion, are nice contributions. If space allows, we will also add and discuss some typical, failed investigations/models we have performed.
>
> [References]
> Bill MacCartney. Natural Language Inference. PhD thesis, Stanford University, 2009.

---

### Official Review · AnonReviewer1 · 2017-11-26
**Alignment is back in NN**

**Rating:** 5
**Confidence:** 4

**Review:**

This is a very interesting paper!
We are finally back to what has been already proven valid for NLI also know as RTE. External knowledge is important to reduce the amount of training material for NLI. When dataset were extremely smaller yet more complex, this fact has been already noticed and reported in many systems. Now, it is extremely important that it is has been started to be true also in NN-based models for NLI/RTE.

Hovever, the paper fails in describing the model with respect to the vast body of research in RTE. In fact, alignment is one of the basis for building RTE systems. Attention models are in fact extremely related to alignment and "KNOWLEDGE-ENRICHED CO-ATTENTION" is somehow a model that incorporates what has been already extensively used to align word pairs.
Hence, models such as those described in the book "Recognizing textual entailment" can be extremely useful in modeling the same features in NN-based models, for example, "A Phrase-Based Alignment Model for Natural Language Inference", EMNLP 2008, or "Measuring the semantic similarity of texts", 2005 or "Learning Shallow Semantic Rules for Textual Entailment", RANLP, 2007.

The final issue is the validity of the initial claim. Is it really the case that external knowledge is useful? It appears that external knowledge is useful only in the case of restricted data (see Figure 3). Hence, it is unclear whether this is useful for the overall set. One of the important question here is then if the knowledge of all the data is in fact replicating the knowledge of wordnet. If this is the case, this may be a major result.

Minor issues
=====
The paper would be easier to read if Fig. 1 were completed with all the mathematical symbols.
For example, where are a^c_i and b^c_i ? Are they in the attention box?

---

> ### Author Response · Authors · 2018-01-05
> **Author Responses**
>
> >>>Reviewer’s comment 1: ”However, the paper fails in describing the model with respect to the vast body of research in RTE”
>
> Author response: The manuscript has focused mainly on neural network based models, but we totally agree with the reviewer--we will add the citations to previous research as kindly suggested by the reviewer. Thank you for the constructive comment.
>
> >>>Reviewer’s comment 2: “...It appears that external knowledge is useful only in the case of restricted data (see Figure 3). ..."
>
> Author response: In the experiments, we found external knowledge constantly improves the performance. As the reviewer pointed out, the improvement is more significant when the size of train data is restricted. It is also significant on when using the entire training set, the proposed model KIM achieved the new state-of-the-art performance on SNLI (88.6% accuracy with a single model and 89.1% with ensembling) over ESIM and the improvement is significant (one-tailed paired t-test at the 99% significance level). We restricted the training data size in order to show the trend and benefit of using external knowledge under different coverage rate. Thank you! We will make these points clearer in revision.
>
>  >>>Minor errors
>
> Thank you so much for pointing out minor errors; we will follow the suggestions to address them in revision.

---

### Official Review · AnonReviewer4 · 2017-12-01
**This paper marginally improves performance on SNLI using a limited set of features indicating WordNet relations. The result is nice but predictable and the methods are not obviously applicable to other external forms of information. This contribution is not sufficient for ICLR.**

**Rating:** 3
**Confidence:** 5

**Review:**

This paper adds WordNet word pair relations to an existing natural language inference model. Synonyms, antonyms, and non-synonymous sister terms in the ontology are represented using indicator features. Hyponymy and hypernymy are represented using path length features. These features are used to modify inter sentence attention, the final post-attention word representations, and the pooling operation used to aggregate the final sentence representations prior to inference. All of these three additions help, especially in the low data learning scenario. When all of the SNLI training data is used this approach adds 0.6% accuracy on the SNLI 3-way classification task.

I think that the integration of structured knowledge representations into neural models is a great avenue of investigation. And I'm glad to see that WordNet helps. But very little was done to investigate different ways in which these data can be integrated. The authors mention work on knowledge base embeddings and there has been plenty of work on learning WordNet embeddings. An obvious avenue of exploration would compare the use of these to the use of the indicator features in this paper. Another avenue of exploration is the integration of more resources such as VerbNet, propbank, WikiData etc. An approach that works with all of these would be much more impressive as it would need to handle a much more diverse feature space than the 4 inter-dependent features introduced here.

Questions for authors:

Is the WordNet hierarchy bounded at a depth of 8? If so please state this and if not, what is the motivation of your hypernymy and hyponymy features?

---

> ### Author Response · Authors · 2018-01-05
> **Author Responses**
>
> Thanks for the constructive comments. We have clarified the questions as follows:
>
> >>> Reviewer’s comment 1: “Is the WordNet hierarchy bounded at a depth of 8? If so please state this and if not, what is the motivation of your hypernymy and hyponymy features?”
>
> Author response: The original WordNet hierarchy is bounded by a depth of 20. In our paper,  we take the setup specified in MacCartney (2009) to bound the depth to 8 (i.e., ignoring pairs in the hierarchy which have more than 8 edges in between).  Similarly, we follow MacCartney (2009) for hypernym and hyponym feature designing, which we show in our experiments to improve neural-network-based inference model to achieve a new state-of-the-art performance (88.6% accuracy with a single model and 89.1% with ensembling).
>
> >>>Reviewer’s comment 2:  "An obvious avenue of exploration would compare the use of these to the use of the indicator features in this paper. "
>
> Thank you. Incorporating WordNet embedding achieved an accuracy of 88.2% on SNLI, compared with 88.6% with KIM and 88.0% with ESIM. WordNet embedding is trained to be sensitive to some semantic relation (e.g., is-a relation which could help detect entailment) but not on others (semantic relations) that would further help NLI (e.g., word pairs with common parents often help identify contradiction). We will add some discussion along this line.
>
> >>>Reviewer’s comment 3: "Another avenue of exploration is the integration of more resources ..."
>
> Author response: WordNet is a lexical and common sense resource that naturally encodes entailment and contradiction information as discussed in the paper and above (e.g., “is-a” and “sibling” relation between word pairs can help resolve entailment and contradiction, respectively). Particularly, considering how NLI data is constructed (e.g., SNLI relies on annotators’ common sense to write entailment and contradiction sentences), we think WordNet is a good resource to demonstrate our algorithms which enhances NN-based NLI models on all three typical NLI submodules/subcomponents. Furthermore, we indeed incorporated WikiData (Freebase) and it did not improve model performance (it is not surprising as most of WikiData is about entities and relations (e.g. Bill Gates and Microsoft) which do not correspond to common entailment/contradiction relation (e.g., red/yellow is contradicting). Thank you for the comment, which we think is very constructive. We will add discussion on this in our revision.
>
> [References]
> Bill MacCartney. Natural Language Inference. PhD thesis, Stanford University, 2009.

---

### Official Review · AnonReviewer3 · 2017-12-01
**Good paper, needs some more experiments.**

**Rating:** 7
**Confidence:** 4

**Review:**

This work is interesting and fairly thorough. The ablation studies at the end of the paper are the most compelling part of the argument, more so than achieving SoTa. Having said that, since their studies on performance with a low dataset size are the most interesting part of the paper, I would have liked to see results on smaller datasets like RTE. Additionally, it would be useful to see results on MultiNLI which is more challenging and spans more domains; using world knowledge with MultiNLI would be a good test of their claims and methods.
I'm also glad that the authors establish statistical significance! I would have liked to see some additional analysis on the kinds of sentences the KIM models succeeds at where their baseline ESIM fails. I think this would be a compelling addition.

Pros:-
- Thorough experimentation with ablation studies; show success of method when using limited training data.
- Establish statistical significance.
- Acheive SoTa on SNLI.

Cons:-
- Authors make the broad claim of world knowledge being helpful for textual entailment, and show usefulness in a limited datasize setting, but don't test their method on other datasets RTE (which has a lot less data). If this helps performance on RTE then this could be a technique for low resource settings.
- No results for MultiNLI shown. MultiNLI has many more domains of text and may benefit more from world knowledge.
- Authors don't provide a list of examples where KIM succeeds and ESIM fails.

---

> ### Author Response · Authors · 2018-01-05
> **Author Responses**
>
> Thank you for the comments. We additionally ran the suggested experiments on the MultiNLI dataset with both the model KIM and ESIM. With the same setting as used in SNLI, KIM achieves a 77.4% accuracy on MultiNLI’s “in-domain” test set (vs. 77.0% of ESIM), and 75.8% on the “cross-domain” test set (vs. 75.5% of ESIM), showing similar and consistent improvement. We will discuss the results in our revision. Thank you. Furthermore, we will include a confusion matrix to detail where KIM corrects the mistakes made by ESIM and include some examples, as suggested by the review.

---

### Public Comment · (anonymous) · 2017-11-07
**Have you tried your method on MultiNLI dataset?**

Good work. I'm wondering whether you have tried your model on MultiNLI corpus? The pipeline should be same. MultiNLI corpus requires higher level understanding of the text. With external knowledge, the performance on MultiNLI should be better than the systems without external knowledge.

---

> ### Author Response · Authors · 2017-11-16
> **MultiNLI results**
>
> We really appreciate the comments. We ran the proposed models on MultiNLI. With the same setting, on the "in-domain" test set, the proposed model, KIM (knowledge-based inference model), achieves an accuracy of 77.4% (vs. 77.0% of the ESIM model (Chen et al. ACL '17)). In addition, on the "cross-domain" test set, KIM achieves a 75.8% accuracy (vs. 75.5% of ESIM). We will add these results in our revision. Thanks for the constructive comments, which make the results more comprehensive.
> [References]
> Q. Chen, X. Zhu, Z. Ling, S. Wei, H. Jiang, and D. Inkpen. (2017). Enhanced LSTM for Natural Language Inference. In: Proc. of ACL, Vancouver, Canada.

---

> > ### Public Comment · (anonymous) · 2017-11-16
> > **Thanks!**
> >
> > It looks great. Thanks!

---

### Decision · Program_Chairs · 2018-01-29
**ICLR 2018 Conference Acceptance Decision**

**Decision:**

Invite to Workshop Track

**Comment:**

the reviewers seem to agree that this submission could be much more strengthened if more investigation is done in two directions: (1) the effect of different, available resources (e.g., in the comment, the authors mentioned WikiData didn't improve, and this raises a question of what kind of properties of external resources are necessary to help) and (2) alternatives to incorporating external knowledge (e.g., as pointed out by one of the reviewers, this is certainly not the only way to do so, and external knowledge has been used by other approaches for RTE earlier. how does this specific way fare against those or other alternatives?) addressing these two points more carefully and thoroughly would make this paper much more appreciated.